# Self-Medication and Safety Profile of Medicines Used among Pregnant Women in a Tertiary Teaching Hospital in Jimma, Ethiopia: A Cross-Sectional Study

**DOI:** 10.3390/ijerph17113993

**Published:** 2020-06-04

**Authors:** Seid Mussa Ahmed, Johanne Sundby, Yesuf Ahmed Aragaw, Fekadu Abebe

**Affiliations:** 1Department of Community Medicine and Global Health, Institute of Health and Society, Faculty of Medicine, University of Oslo, P.O. Box 1130 Blindern, N-0318 Oslo, Norway; johanne.sundby@medisin.uio.no (J.S.); fekadu.abebe@medisin.uio.no (F.A.); 2Division of Social and Administrative Pharmacy, School of Pharmacy, Faculty of Health Sciences, Jimma Institute of Health, Jimma University, Jimma P.O. Box 378, Ethiopia; 3Department of Obstetrics and Gynaecology, Faculty of Medical Sciences, Jimma Institute of Health, Jimma University, Jimma P.O. Box 378, Ethiopia; yesufahmed47@yahoo.com

**Keywords:** pregnancy, self-medication, conventional medicine, safety, Ethiopia

## Abstract

Background: Despite the potential foetal and maternal risks of self-medication, studies on self-medication and safety profile of medicines used during pregnancy are scarce. This study determined the prevalence, predictors and safety profile of medicines used for self-medication during pregnancy at Jimma University Medical Centre (JUMC) in Ethiopia. Methods: A hospital-based cross sectional study was conducted on 1117 hospitalized pregnant women or postpartum women in the maternity and gynaecology wards at JUMC between February and June 2017. Data were collected using an interviewer-administered structured questionnaire and by reviewing patient medical records. Data were analysed using descriptive statistics and logistic regression. Result: Nearly 3 out of 10 women reported taking at least one type of conventional medicine for self-medication, mainly analgesics 92.3%. Almost 75.0% of the self-medicated women used medicines classified as probably safe and 13.6% as potentially risky to use during pregnancy. Medicinal plant use, religion and access to a health facility near their residency were significantly associated with self-medication during pregnancy. Conclusions: Self-medication is common among pregnant women at JUMC. Most women used medicines classified as safe to use during pregnancy. There is need for enlightenment of pregnant women on the potential dangers of self-medication during pregnancy to prevent foetal and maternal risks.

## 1. Introduction

Pregnancy is a dynamic process in which many maternal physical, physiological, pharmacokinetic and pharmacodynamic transformations occur from fertilization to parturition [1,2,3]. Changing hormones can alter a woman’s mood and cause nausea, vomiting, heartburn, constipation, headache, cough and other weakening pregnancy pains that force expectant mothers to seek healthcare [2,3,4,5]. However, in most developing countries like Ethiopia not only are health care facilities inaccessible or unaffordable [2,4,6], but also medicines are poorly regulated and easily available outside formal and authorized institutions [7]. For this reason, many pregnant women prefer to self-medicate first as an accessible and lower cost alternative and only seek professional health services when the situation worsens [4,8].

Self-medication is defined as the use of medicines or the intermittent or continued use of prescribed medicines by the patient to treat self-diagnosed disorders or symptoms on their own initiative [8,9]. What makes self-medication more dangerous in developing countries is that the population has low health literacy, and thus may have a poorer understanding of health information and inappropriate use of medicines [8,10]. In Ethiopia, the prevalence of self-medication with conventional medicines in pregnancy has been shown to vary from 1.9% [11] to 29.1% [12]. Patients self-medicating with western medicine may use medicinal plants concomitantly and often without the knowledge of a healthcare professional, which might further threaten the pregnancy [2,13]. Furthermore, medicines taken by the mother may cause serious structural and functional adverse effects on the developing baby including developmental delay, foetal toxicity, low birth weight, intellectual disability, birth defects, miscarriage and stillbirth [7,14].

Self-medication in pregnancy is common in Sub-Saharan African countries, potentially exposing the unborn child to medicines that may have risk [15]. Potentially risky medicine use in pregnancy has been reported, with prevalence estimates of 28.0% in Europe [16] and 49.5% in Burkina Faso [15].

Different classification systems of risks of medicines used during pregnancy have been established depending on foetal safety. The most well-known are the U.S., the Australian and the Swedish pregnancy risk classification systems [17]. Despite their limitations, these systems are important in describing medicine utilization patterns [16].

In addition to self-medication, pregnant women may be unaware that social drug use could affect the health of the foetus. Substance use during pregnancy has been associated with miscarriage, stillbirth, preterm birth, restricted foetal growth, low birth weight, babies who are small for their gestational age and intellectual impairment in children and can contribute to maternal or delivery complications [3,18,19]. Smoking as high as 5.0% [20], alcohol consumption as high as 39.8% [18], and khat (*Catha edulis*) chewing varying from 10.0% [21] to 35.8% [22] has been reported in pregnant women in Ethiopia.

The government of Ethiopia is working hard to achieve Sustainable Development Goal 3 (SDG 3) to reduce the maternal mortality ratio (MMR) from 412 deaths per 100,000 live births in 2015 to less than 70 deaths per 100,000 live births by 2030 [23]. Identifying the extent and determinants of self-medication, safety profile of medicines used for self-medication and social drugs used could help in educating and counselling pregnant women and in achieving the SDG 3 targets. However, in Ethiopia, despite the fact that its negative pregnancy outcomes pose a significant threat to maternal and foetal health [24], data systematically examining prevalence and correlates of self-medication in pregnancy are still scarce [7,12,25,26]. Moreover, the limited self-medication studies performed in the country have been confined to outpatient pregnant women [7,12]. Since we did not find any study that assessed self-medication experience and safety profiles of medicines used among hospitalized pregnant women, this study aimed to determine the magnitude of self-medication practice and its determinants among pregnant women at JUMC. The study also evaluated pregnancy safety profiles of medicines used based on risk classification techniques developed in the U.S. and Australia. A secondary aim was to assess the prevalence and types of social drugs used among the pregnant women at JUMC.

## 2. Subjects and Methods

### 2.1. Study Design and Setting

A facility based cross-sectional study was conducted in the maternity and gynaecology wards of a tertiary care teaching hospital, JUMC, Ethiopia. Geographically, the hospital is located in Jimma city 350 km southwest of the capital city of Ethiopia, Addis Ababa. It is one of the oldest public hospitals in the country established in 1937/38 [27]. Currently it is the only teaching and referral hospital in the south-western part of the country, with a catchment population of about 20 million people [28]. The JUMC obstetrics and gynaecology department has a bed capacity of 265 and provides specialized health services for about 7600 inpatients and 11,600 outpatients every year.

The department of obstetrics and gynaecology has two wards (maternity and gynaecology), one antenatal care (ANC) outpatient clinic, one general gynaecological outpatient clinic and one family planning clinic [28]. Women with a gestational length of 28 weeks or higher and women in labour receive care in the maternity ward. Women with less than 28 weeks of pregnancy (most often hyperemesis and abortions) are treated at the gynaecology ward. The gynaecology ward also manages and treats gynaecological conditions in non-pregnant patients. Ethics approval and consent to participate: The study was approved by the Regional Committees for Medical and Health Research Ethics (REC) in Norway (Ref.: 2015/2135, REK Sør-Øst B), dated 17 December 2015 and the Institutional Review Board (IRB), Institute of Health, Jimma University, Ethiopia (ref. no IHRPGC 7206/07), dated 17 January 2017. Permission was secured from JUMC before commencing the study. Written informed consent was obtained from each study participant before data collection. All information obtained from participants during the study was kept confidential.

### 2.2. Study Population and Sample Size

Pregnant or postpartum women in the maternity and gynaecology wards at JUMC were invited to participate in the study during their inpatient hospital stay. Since there are no previous studies that reported the prevalence of self-medication with conventional medicine among pregnant women prior to hospital admission, we used 50% as a conservation estimate. The sample size was based on having a power of 80%, a critical level of significance of 5% and an error margin of 3% using the Kish single population formula [29] provided below in Equation (1). Sample size calculation was based on the study’s primary objective, that is, to provide an estimate of self-care with conventional medicines among pregnant women prior to hospitalization.
(1)$$N=Z_{\alpha /2}^{2}\times \frac{p\left(1-p\right)}{\varepsilon^{2}}=1.96^{2}\times \frac{0.5\left(1-0.5\right)}{0.03^{2}}=1067$$
*N* is the sample size; *Z* is the standard normal deviate (the *Z* value for 95% confidence level is 1.96).

We also allowed the possibility of a 5% non-response rate (approximately 54 women). Therefore, at least 1121 women were required for the study to have enough power.

### 2.3. Inclusion and Exclusion Criteria

Inclusion criteria: Pregnant or postpartum patients aged ≥18 years admitted in the maternity/labour and gynaecology wards at the time of data collection and willing to participate in the study.

Exclusion criteria: Women whose physical and psychological health limited them from providing information, such as those who were unable to speak or mentally disabled, too ill to participate or hard of hearing, were excluded from the study. In addition, women who were unwilling to participate, admitted for less than four hours, under 18 years of age and non-pregnant women admitted in the gynaecology ward were excluded.

### 2.4. Data Collection

Data were collected using a pre-tested face-to-face interviewer-administered structured questionnaire and by reviewing patient medical records. A patient chart review was used to collect pregnancy characteristics, pregnancy outcomes and other medical information about pregnant women. Before the interviews, the aims, objectives and procedures of the study were clearly explained to the participants. After securing written informed consent from each hospitalized pregnant or post-partum woman, the women were consecutively interviewed from February to June 2017. In addition, data were collected at an appropriate and convenient time for the women. Nine trained data collectors, five clinical pharmacists and four nurses from JUMC collected the data. They were given training on how to interview patients using the questionnaire and verify the completeness of the filled questionnaire and abstract information from patient medical records. One of the investigators supervised the data collection and verified the completeness of each questionnaire every day. To ensure confidentiality, the questionnaire did not include the woman’s name or any other identifying information.

### 2.5. Development of Data Collection Tool

The bilingual questionnaire was developed based on a review of relevant literatures. It was initially developed in English, then translated into the local languages, Amharic and Afan Oromo, and back into English to ensure consistency. The data collection tool was pre-tested on a sample of 30 inpatient pregnant or post-partum women at Shenen Ghibe district hospital located in Jimma city, to assess content validity, content consistency, comprehension and possible defective questions and the time needed to complete it. Based on the pre-test, the questionnaire was amended accordingly and data collectors were clarified on items which were not understood well. The data extraction form was single page and required only minor amendments.

The questionnaire was comprised of four sections to address the aims of this study. Section 1 contained questions about the women’s socio-demographic characteristics including age, religion, place of residence, occupation, family size, ethnic group, marital status, educational level and access to a modern health facility. Section 2 contained questions about history of maternal medical problems and maternal and perinatal outcomes. Pregnant women were asked specifically about medical history, pregnancy illnesses and known chronic diseases. Section 3 contained questions about self-medication practice. Self-medication practice was assessed by asking women to list any medication they used by themselves including medications leftover from previous facility visits, bought without a prescription paper from drug retail outlets or shared by anyone and used for the management of their illnesses. Participants were also asked to provide names for any supplement or preparation they may have taken including iron, folic acid and any other supplement. Section 4 covered social drug use history, particularly tobacco smoking, alcohol drinking and *khat* chewing during pregnancy. To investigate the use of social drugs, women were asked if they had used any of the listed social drugs—tobacco, alcohol and *khat*. Moreover, participants were asked to indicate any other social drug used, the amount and the duration of use.

In addition to the questionnaire, a data extraction form was used to collect information about pregnancy characteristics, pregnancy outcomes and other obstetrics information including parity, gravidity, gestational age, delivery route and length of hospital stay. Moreover, maternal and perinatal outcomes of the current pregnancy were collected. Data were extracted by reviewing patients’ medical cards.

In this study, concomitant use of medicinal plants and pharmaceutical medicines was assessed by identifying those women who used both during pregnancy for the same or different illnesses.

### 2.6. Safety Classification of Medicines

To attribute each medicine in risk groups according to foetal safety, medicines were classified using two globally recognised risk classification systems commonly used in Ethiopia, the U.S. Food and Drug Administration (US FDA) and the Australia Therapeutic Goods Administration (AU-TGA). The FDA classification system, which uses five categories, A, B, C, D and X [30] was used as the primary categorization approach because it is widely used in Ethiopia. The FDA Category A indicates the safest medications, whereas category X designates medications that have been shown to be teratogenic. The FDA amended their pregnancy risk letter categories in June 2015 and this type of categorization is no longer used [30]. However, it was not only in use during this study but also still widely used in Ethiopia [7,11]. If a particular medicine was not covered by the FDA classification, the AU-TGA classification system [31] was used as a secondary method of classification. It has classes (A, B1, B2, B3, C, D and X) to define medicine safety. Based on similar previous studies [15,16], in order to facilitate the safety analysis and to make categories of more clinical interest, medicine exposures were classified into “probably safe”, “potentially risky” or “unclassified”. For pharmaceuticals manufactured with several active ingredients; the risk classification was done based on the active ingredient with the highest risk. Similarly, for combination medicines, risk class was assigned based on the dominant active substance. According to these two classifications, the “probably safe” medicines group consisted of the FDA categories A and B, and the AU-TGA categories A, B1 and B2. Categories C, D and X for FDA and categories B3, C, D and X for AU-TGA are classified as “potentially risky”. Medicines that could not be classified by any of these resources were registered as “unclassified”. Finally, when necessary, findings from the FDA or AU-TGA groupings were modified guided by the Ethiopian epidemiological profile, national formulary and treatment guidelines for disease treatment and the WHO recommendations.

### 2.7. Statistical Analysis

One of the investigators (SMA) verified the filled-in questionnaire for completeness and consistency and then coded, entered, cleaned and finally analysed using Statistical Package for the Social Sciences (SPSS) software version 25.0 for Windows (IBM^®^ SPSS^®^ Statistics for Windows, Version 25.0, IBM Corp, Armonk, NY, USA). Descriptive statistics were used to summarize the data at baseline. To identify independent factors significantly associated with self-medication, univariate and multivariate logistic regression analyses were computed and expressed as crude and adjusted odds ratios (ORs) with 95% confidence intervals (CIs). Independent variables with *p* < 0.25 in a univariate logistic regression model were fit into a multivariate model to determine predictors of self-medication. Significance was set at the standard alpha of 0.05. Whenever the *p*-value was found to be <0.05, the association was considered statistically significant. Similar data processing and analysis procedures were used for social drugs.

## 3. Results

### 3.1. Study Population Characteristics

Out of the 1137 pregnant and nursing women invited to participate in the study, 1121 of them agreed to take part, making the response rate 98.6%. However, responses from four women were incomplete and thus complete data on 1117 women (18–45 years old, with a median age 25 years) were analysed. Most of the study participants were urban inhabitants (53.3%), housewives (46.9%) and had attended primary school or were able to read and write (42.3%). Around two-thirds were from Oromo ethnic group (69.7%), professed the religion of Islam (65.4%), had a household size less than five (65.4%) and lived in a place within walking distance of less than half an hour from the nearest health facility (66.4%). The majority were married women (96.3%) and admitted in the maternity ward (88.8%) and over half of them (54.7%) delivered vaginally. Five percent of the participants had known chronic disease. The frequencies and percentages of sociodemographic and pregnancy related characteristics of participants are presented in Table 1.

### 3.2. Self-Medication Practice

Table 2 depicts the medicines used for self-medication, the Anatomical Therapeutic Chemical Classification System (ATC) code and the medicine safety category. Of all the pregnant women surveyed, (27.0%) reported taking at least one type of conventional medicine for self-medication, mainly analgesics (92.3%) followed by antibiotics (6.7%) and gastrointestinal (GI) medicines (4.3%). With regard to specific medicines, paracetamol (72.7%) followed by diclofenac (11.0%) and amoxicillin (5.7%) were the most commonly used medicines for self-medication. Nearly a third of women (28.6%) had used one or more medicinal plant during their current pregnancy. Moreover, 110 (36.7%) of the self-medicated women had also used one or more medicinal plants along with the conventional medicines during pregnancy. The women used 1 to 8 types of medicinal plants, with an average of 1.63 plants per woman. The majority of these women, 67 (60.9%), used one medicinal plant, whereas 27 (24.5%) used two and 12 (10.9%) used three. Moreover, two women used four, one woman used six and another woman used eight medicinal plants along with self-medication during pregnancy. *Telba* (Flaxseed) 88 (80.0%) was the most commonly used medicinal plant followed by *Zingibil* (Ginger) 15 (13.6%) (Appendix A).

### 3.3. Factors Associated with Self-Medication Practice during Pregnancy

Women who used medicinal plants in the current pregnancy (adjusted OR 1.78; 95% CI 1.33, 2.40) and Islam (adjusted OR 2.22; 95% CI 1.19, 4.17) or Orthodox Christian (adjusted OR 2.04; 95% CI 1.06, 3.92) religion followers were more likely to practice self-medication during pregnancy than Protestant Christians and other religious groups. On the other hand, women who had access to a health facility near their place of residence (adjusted OR 0.62; 95% CI 0.41, 0.95) were less likely to employ self-medication practice during pregnancy (*p* < 0.05). In univariate analysis, self-medication practice during pregnancy was also strongly associated with educational level, obstetrics category and patient type; however, this was not maintained in multivariable logistic regression analysis (Table 1). Other characteristics and pregnancy outcomes like gravidity, parity, gestational age, patient type, delivery route (obstetrics category) and length of hospital stay were not associated with self-medication.

### 3.4. Summary of Safety Classification of Medicines Used in Pregnancy

As shown in Table 2, using the US FDA/AU-TGA classification method, 243 (73.4%) of the pharmaceuticals used for self-medication were classified as probably safe to use during pregnancy, most commonly paracetamol 218 (65.9%). A total of 45 (13.6%) medicines were classified as potentially risky to use during pregnancy, mainly diclofenac 33 (10.0%), followed by ibuprofen 9 (2.7%). While there was no medicine in which no classification was available, there were 43 (13.0%) safety ‘‘undetermined’’ medicines because women did not remember the names of the medicines used.

### 3.5. Social Drugs Used among Pregnant Women

The prevalence of at least one substance use among pregnant women (either *khat* chewing or alcohol drinking) was 9.7%. Around five percent of the pregnant women consumed alcohol during the current pregnancy, mainly local beer (73.9%). Surprisingly 31 women daily drank variable amounts of alcohol. The proportion of *khat* chewers was around 6.0%, and 97.0% of them chewed daily and almost half of them chewed over 166.0 g of *khat* every day. None of the pregnant women was an active tobacco smoker (Table 3).

### 3.6. Factors Associated with Use of Social Drugs during Pregnancy

Women who used medicinal plants during pregnancy (adjusted OR 2.75; 95% CI 1.79, 4.24), women admitted to the gynaecology ward (adjusted OR 2.81; 95% CI 1. 31, 6.04) and Islam (adjusted OR 3.79; 95% CI 1.13, 12.76) or Orthodox Christian (adjusted OR 4.08; 95% CI 1.19, 13.99) religion followers were more likely to have used social drugs during pregnancy than their counterparts. On the other hand, women who had shorter hospital stays were less likely to have used social drugs during pregnancy (adjusted OR 0.63; 95% CI 0.41, 0.95) (*p* < 0.05) (Appendix A).

## 4. Discussion

To the best of our knowledge, this is the first study to investigate self-medication and the safety profiles of medicines used during pregnancy among hospitalized women using both the US FDA and the AU-TGA pregnancy risk classification systems in Ethiopia. This study found that over a quarter (27.0%) of mothers self-medicated with at least one type of conventional medicine, mostly analgesics (92.7%), at some stage of the current pregnancy. A very concerning finding is that (6.7%) of the self-medications were with antibacterials. Moreover, (36.7%) of the self-medicated women took at least one medicinal plant concomitantly during pregnancy. Self-medication among pregnant women was found to be influenced by several factors such as walking distance to the nearest health facility, medicinal plant use and religion. Besides, one in ten of the study participants used at least one social drug in pregnancy.

The self-medication rate in our study was comparable to some previous studies [12,26] but larger than other studies [7,11,25] from Ethiopia. This discrepancy in prevalence could emanate from different reasons. First, in this study women were requested to recall self-medication over a nine month period, whereas in many other studies the recall period was limited to two weeks. Second, participants in this study were hospitalized women (most in their third trimester), whereas the women in previous studies were outpatients (most in early stage of pregnancy). Third, the two previous studies with lower prevalence [7,11] were from the capital of Ethiopia, Addis Ababa, which is supposed to have better access to modern health care services and better knowledge, resulting in reduced self-medication rate, compared to its rural counterparts. Finally, the variation in the study settings, methodology, restriction policies on dispensing medicines and disease distribution may partly explain the disparity in the reported magnitude of self-medication.

The worrisome finding in this study is that almost one in fifteen women were using antimicrobials for self-medication. Antibiotic self-medications have been reported in many parts of Africa, Asia, Europe, North and South America [32] at varying rates showing that self-medication with antimicrobials is a global challenge. While responsible self-medication is an important response to medical conditions, self-treatment with antimicrobials is associated with the risk of irrational medicine use, which predisposes patients to antimicrobial resistance and shrinks the range of effective antimicrobials [14,32]. In Ethiopia, where there is low literacy, weak regulation of use of medicine, weak enforcement of regulation and less strict dispensing policies, the dangers posed by antimicrobial resistance to the gestating women, the foetus and the society are immense, demonstrating that rational medicine use practices should be a priority agenda.

Owing to the large quantity of compounds present in medicinal plants, on top of potential inherent toxicity, many different interactions can occur with concomitant use of other pharmaceuticals, which further jeopardizes maternal and foetal safety [33]. The reported rate of concomitant use of medicinal plants with pharmaceuticals during pregnancy in this study (10.0%) falls within the range 2.4–77.3% reported in literature among African pregnant women [2]. Methodological inconsistencies in capturing concomitant use and variations in the study setting, study participants and the definition of what “medicinal plant” encompasses may be the reasons for the differences. As we could only describe co-use and not assess interactions in the current study, there is still a need for future research on the potential clinical implications of interactions for women using both treatment modalities in pregnancy.

Women who used medicinal plants during the current pregnancy were almost twice as likely to self-medicate sometime during pregnancy than non-users. This corroborates previous studies [2,8], which claimed that many pregnant women self-care with medicinal plants first and only seek professional health services when the situation worsens. Such customs may delay proper medical care seeking and possible interaction between plant remedies and conventional medicine, which could be detrimental to maternal and neonatal health.

Gestating women who had a health facility in their neighbourhood were 38% less likely to self-medicate during pregnancy. The identification of a nearby location of health facilities as a key factor discouraging self-medication during pregnancy may be particularly important in urban and peri-urban areas of Ethiopia (like in our study in which more than half were urban residents), where functional access roads and transport services are often in better supply [3,34]. A sizeable proportion of pregnant women may therefore seek proper medical care at nearby health facilities rather than resorting to self-medication to meet their perceived health needs.

In agreement with a study from Eritrea [35] a significant difference was observed in self-medication practice among religious groups, for no known reason. Thus, a topic for future research could be to uncover any underlying factors.

Compared to previous studies [15,16], it is reassuring that a significantly larger proportion of women used medicines categorized as safe to use in pregnancy. On the other hand, 13.6% of the self-medicated women used potentially risky medications; however, this result is still promising and lower than findings from previous studies [7,15,16]. Differences in use of medicine between studies with respect to safety classification may be attributed to disparities in the study design, geography, variables used and size, pregnancy type, demographic characteristics, variety of medicines on the market and their availability to pregnant women. Nevertheless, the results of the present study still revealed that segments of women were exposed to potentially risky medicines, warranting closer attention.

Concerning the safety of medicinal plants used by women, research indicates that flaxseed use in the last two trimesters of pregnancy is associated with increased risk of premature birth, warranting cautious use [36]. On the other hand, *Zingibil* (Ginger) use is not known to have detrimental effects on the foetus [2].

Our study disclosed that almost all chewers chewed at least 125.0 g *khat* daily and had been chewing for a long time (Table 3). As *khat* has a known appetite suppression effect [37,38], it could lead to a poor nutritional status of the mother and inadequate weight gain during pregnancy, resulting in low birth weight. In addition, *khat* is associated with constipation [37]; coupled with the constipation effect of iron supplement, *khat* may hamper treatment adherence and further jeopardize the health of the foetus and the mother. Considering its harmful health impact, *khat* chewing should be discouraged in pregnancy through appropriate educational intervention.

### Study Strengths and Limitations

This study has several strengths. The main strength is the large sample size of 1117 women that provides greater insight into the used medicines that might have the potential for foetal harm. Equally, health professionals in the study area with knowledge about the healthcare system, local language, culture and previous research or practice experience collected the data. This study had some limitations as well. It was conducted in a tertiary care hospital in Ethiopia and, therefore, this may not be representative of the pregnant female population who accessed primary or secondary care services. Medicine exposure information was collected by relying on the women’s recall from their entire pregnancy; therefore, there is a possibility of recall bias leading to underestimation of medicine use among women. Another limitation is that the women may not remember the names of medicines used, which obstructs safety class categorization. Additionally, aborting women were included in the study and their decision to receive an abortion may have influenced their self-medication selection. Finally, since the study was institutional, pregnant women might be confused or embarrassed about disclosing their use of medicines.

## 5. Conclusions

Overall, approximately three out of ten pregnant women self-medicated with at least one type of conventional medicine, mainly analgesics, in this Ethiopian setting. It is reassuring that the majority of women self-medicated with medicines probably safe to use during pregnancy. However, it is also concerning that some women used potentially risky medicines or concomitantly self-cared with medicinal plants. Access to health facility, medicinal plant use and Islam or Orthodox Christian religion were factors associated with self-medication. The findings suggest that there is need for educating pregnant women on the types of illnesses that can be self-diagnosed and self-treated, and the types of medicines to be used for self-care in pregnancy to promote responsible self-medication and prevent foetal and maternal risks. In addition, there is a need to raise awareness among women of the fact that even over-the-counter medicines do require advice and counselling from health personnel. Finally, health facilities also need to routinely include pregnant women self-medication history in patient medication records to prevent potential harms of self-medication.

## Figures and Tables

**Table 1 ijerph-17-03993-t001:** Characteristics of pregnant women according to self-medication practice prior to admission to JUMC, Ethiopia.

Characteristics	No. (%) 1117 (100) ^a^	Self-Medication Practice ^b^		Crude OR [95% CI] ^c^	Adjusted OR [95% CI] ^d^
		No	Yes		
		No. (%) 817 (73.1)	No. (%) 300 (26.9)		
**Place of Residence**					
Urban	595 (53.3)	431 (52.8)	164 (54.7)	1.08 [0.83–1.41]	
Rural	522 (46.7)	386 (47.2)	136 (45.3)	1	-
**Age (years)** ^e^					
≤20	223 (20.0)	158 (19.3)	65 (21.6)	1.25 [0.81–1.95]	
21–25	388 (34.7)	285 (34.9)	103 (34.3)	1.10 [0.74–1.64]	
26–30	320 (28.7)	234 (28.6)	86 (28.7)	1.12 [0.74–1.69]	
≥31	186 (16.7)	140 (17.1)	46 (15.3)	1	-
**Marital status**					
Married	1071 (95.9)	787 (96.3)	284 (94.7)	0.68 [0.36–1.26]	0.62 [0.31–1.24]
Others ^f^	46(4.1)	30 (3.7)	16 (5.3)	1	1
**Religion**					
Islam	731 (65.4)	533 (65.2)	198 (66.0)	1.78 [0.98–3.24]	**2.22 [1.19–4.17]**
Orthodox Christian	305 (27.3)	217 (26.6)	88 (29.3)	**1.94 [1.04–3.63]**	**2.04 [1.06–3.92]**
Protestant/Others ^g^	81 (7.3)	67 (8.2)	14 (4.7)	1	1
**Educational level** ^h^					
Illiterate	378 (34.0)	285 (34.9)	93 (31.0)	0.68 [0.42–1.10]	0.60 [0.36–1.00]
Primary/read & write	470 (42.3)	353 (43.2)	117 (39.0)	0.69 [0.44–1.10]	0.65 [0.40–1.05]
Secondary school	162 (14.6)	107 (13.1)	55 (18.3)	1.08 [0.63–1.82]	0.93 [0.54–1.60]
Post-secondary school	102 (9.2)	69 (8.4)	33 (11.0)	1	1
**Occupation**					
Housewife	524 (46.9)	396 (48.5)	128 (42.7)	0.87 [0.50–1.51]	
Farmer	261 (23.4)	190 (23.3)	71 (23.7)	1.01 [0.56–1.80]	
Trader/Merchant	163 (14.6)	112 (13.7)	51 (17.0)	1.23 [0.67–2.26]	
Government employee	95 (8.5)	65 (8.0)	30 (10.0)	1.25 [0.64–2.44]	
Others ^i^	74 (6.6)	54 (6.6)	20 (6.7)	1	-
**Ethnic Group**					
Oromo	779 (69.7)	573 (70.1)	206 (68.7)	1.02 [0.64–1.64]	
Amhara	87 (7.8)	60 (7.3)	27 (9.0)	1.28 [0.68–2.42]	
Yem	81 (7.3)	55 (6.7)	26 (8.7)	1.35 [0.71–2.57]	
Dawuro	70 (6.3)	55 (6.7)	15 (5.0)	0.78 [0.38–1.60]	
Others ^j^	100 (9.0)	74 (9.1)	26 (8.7)	1	-
**Walking distance to the nearest health facility**					
Close, ≤30 min.					
Somewhat far, 31–60 min.	731 (66.4)	542 (66.3)	189 (63.0)	0.71 [0.48–1.08]	**0.62 [0.41–0.95]**
Far, >60 min.	245 (22.3)	177 (21.7)	68 (22.7)	0.79 [0.49–1.26]	0.74 [0.46–1.19]
**Gravidity** ^k^	125 (11.4)	84 (10.3)	41 (13.7)	1	1
Primigravida					
Multigravida	431 (38.6)	307 (37.6)	124 (41.3)	1.17 [0.89–1.53]	
**Gestational age**	686 (61.4)	510 (62.4)	176 (58.7)	1	-
Preterm pregnancy					
Term pregnancy	231 (20.7)	157 (19.2)	74 (24.7)	1.53 [0.87–2.68]	1.36 [0.71–2.62]
Post term pregnancy	735 (65.8)	542 (66.3)	193 (64.3)	1.15 [0.68–1.93]	1.17 [0.69–1.99]
Others	62 (5.6)	50 (6.1)	12 (4.0)	0.78 [0.35–1.73]	0.69 [0.30–1.57]
**Patient type**	89 (8.0)	68 (8.3)	21 (7.0)	1	1
Gynaecology ward					
Maternity ward	125 (11.2)	81 (9.9)	44 (14.7)	**1.56 [1.05–2.32]**	2.18 [0.41–11.51]
**Chronic illness** ^l^	992 (88.8)	736 (90.1)	256 (85.3)	1	1
Yes					
No	56 (5.0)	39 (4.8)	17 (5.7)	1.20 [0.67–2.15]	
**Medicinal plant use**	1061 (95.0)	778 (95.2)	283 (94.3)	1	-
Yes					
No	319 (28.6)	209 (25.6)	110 (36.7)	**1.68 [1.27–2.23]**	**1.78 [1.33–2.40]**
**Chew *khat* (*Catha edulis*)** ^m^	798 (71.4)	608 (74.4)	190 (63.3)	1	1
Yes					
No	1052 (94.2)	47 (5.8)	18 (6.0)	1.05 [0.60–1.83]	
**Alcohol consumption**	65 (5.8)	770 (94.2)	282 (94.0)	1	-
Yes					
No	46 (95.9)	30 (3.7)	16 (5.3)	1.48 [0.79–2.75]	1.31 [0.67–2.59]
	1071 (4.1)	787 (96.3)	284 (94.7)	1	1
**Obstetrics Category**					
Caesarean Delivery(CD)	372 (33.3)	276 (33.8)	96 (32.0)	0.67 [0.44–1.02]	1.74 [0.33–9.29]
Vaginal delivery(VD)	611 (54.7)	453 (55.4)	158 (52.7)	**0.67 [0.45–0.99]**	1.69 [0.32–8.92]
Antenatal or others ^n^	134 (12.0)	88 (10.8)	46 (15.3)	1	1

^a^ Numbers may not add up to 1117 due to missing values; ^b^ Self-medication practice with conventional medicines before hospitalization; ^c^ CI, confidence interval, OR, odds ratio; Significant findings are in bold (*p* < 0.05); ^d^ Adjusted for marital status, religion, educational level, walking distance to the nearest health facility, gestational age, patient type, medicinal plant use, alcohol consumption, obstetrics category; ^e^ Median age 25 years, interquartile range 22–30 years; ^f^ Others includes single 41 (3.7%), divorced 4 (0.4%), widowed 1 (0.1%); ^g^ Protestant/Others includes Protestant 74 (6.6), Catholic 2 (0.2%), Waqqefeta 1 (0.1%), missing 4 (0.4); ^h^ Read & write: no formal education but can read and write due to literacy campaigns, traditional religious institution and informal peer learning, Primary school: Grade 1–8, Secondary school: Grade 9–12; Post-secondary school: Technical and vocational school, college, university; ^i^ Others includes daily labourers 24 (2.1), students 22 (2.0), private institution workers 18 (1.6), other sectors 10 (0.9%); ^j^ Others includes Silte 30 (2.7), Kaffa 16 (1.4), Tigre 3 (0.3), Wolayita 3 (0.3), mixed ethnic backgrounds 7 (0.6); ^k^ Gravidity includes the current pregnancy; ^l^ includes asthma, cardiac diseases, chronic gastritis/peptic ulcer, hypertension, HIV, chronic renal failure, chronic liver disease, diabetes mellitus, etc. ^m^
*khat* (*Catha edulis*) plant leaves are chewed by people for their stimulant action; ^n^ Others includes pregnant women admitted in the Gynaecology ward due to elective and/or spontaneous abortions, hyperemesis gravidarum or other with early pregnancy health concerns.

**Table 2 ijerph-17-03993-t002:** Self-medication practice among pregnant women prior to Jimma University Medical Center (JUMC) admission and modified medicine risk classification, Ethiopia, February to June 2017.

Therapeutic Class and INN (*N* = 300)	*n* (%) ^a^	ATC Code	Risk Category ^b^
**Analgesics** (*N* = 277)			
Paracetamol	218 (72.7)	N02BE01	Probably safe
Diclofenac	33 (11.0)	M01AB05	Potentially risky
Antipain medicines ^c^	26 (8.7)	-	Undetermined
Ibuprofen	9 (3.0)	M01AE01	Potentially risky
Tramadol	2 (0.7)	N02AX02	Probably safe
**Antibacterial** (*N* = 20)			
Amoxicillin	17 (5.7)	J01CA04	Probably safe
Cloxacillin	1 (0.3)	J01CF02	Probably safe
Metronidazole	1 (0.3)	J01XD01	Probably safe
Antibacterial medicines ^c^	1 (0.3)	-	Undetermined
**GI medicines** (*N* = 13)			
Antacid ^c^	10 (3.3)	-	Undetermined
Omeprazole	1 (0.3)	A02BC01	Potentially risky
Metoclopramide	1 (0.3)	A03FA01	Probably safe
Hyoscine butylbromide	1 (0.3)	A03BB01	Probably safe
Magnesium sulphate	1 (0.3)	A06AD04	Potentially risky
**Antihelmenthics** (*N* = 3)			
Anthelminthic medicines ^c^	3 (1.0)	-	Undetermined
**Supplements** (*N* = 2) ^d^			
Ferrous sulphate	1 (0.3)	B03AA07	Probably safe
Multivitamin Tablets	1 (0.3)	A11AA05	Probably safe
**Antihypertensives** (*N* = 1)			
Hydralazine	1 (0.3)	C02DB02	Potentially risky
**Other classes** (*N* = 3)			
Name forgotten Medicines ^e^	3 (1.0)	-	Undetermined

^a^ Percentage may exceed 100% due to multiple responses, and Percentage is calculated taking those who self-medicated with conventional medicine as a denominator, *N* = 300. ^b^ Medicine safety classification was based on modified US-FDA and AU-TGA risk category; In addition to the three modified safety classes, a fourth class ‘‘undetermined’’ was added for medicines whose exact names the women did not remember. ^c^ The women did not remember the exact names of each medicine. ^d^ In Ethiopia, most supplements are considered as medicines and are included in the List of Drugs for Ethiopia (LIDE), 5th edn., 2007. ^e^ Women used a medicine but did not remember either the reason for use or the exact names of each medicine. Abbreviations: AU-TGA Australian Therapeutic Goods Administration, US FDA United States Food and Drug Administration, ATC Anatomical Therapeutic Chemical, INN International Non-proprietary Name.

**Table 3 ijerph-17-03993-t003:** Social drugs used among pregnant women prior to admission to JUMC, Ethiopia, February to June 2017.

Characteristics	Frequency	Percent ^a^
**Smoke tobacco** (*N* = 1117)		
Yes	0	0
No	1117	100
**Drink alcohol** (*N* = 1117)		
Yes	46	4.1
No	1071	95.9
**Type of alcohol** (*N* = 46)		
Tella (Local beer)	34	73.9
Beer	12	26.1
Wine	7	15.2
Katikala (‘Ethiopian vodka’)	4	8.7
Type not given/indicated	1	2.2
**Amount of alcohol consumed** (*N* = 46)		
Regular or daily use of variable amounts	31	67.4
Irregular use of variable amounts	15	32.6
**Chew *khat*** (*N* = 1117)		
Yes	65	5.8
No	1052	94.2
**Length of years chewed** (*N* = 65)		
≤5	29	44.6
>5	36	55.4
**Amount chewed daily** (*N* = 65) ^b^		
One zurba	21	32.3
Two thirds of a zurba	11	16.9
≤One half of a zurba ^c^	33	50.8

^a^ Percentage may exceed 100% due to multiple responses, ^b^ Women reported in terms of a local measurement unit, the “zurba”; the approximate weight of one zurba *khat* plant leaf ≈ 250 g. ^c^ Two of them chewed varying amounts and infrequently.

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
