# Peer review of "Self-Medication and Safety Profile of Medicines Used among Pregnant Women in a Tertiary Teaching Hospital in Jimma, Ethiopia: A Cross-Sectional Study"

_ijerph, 2020, doi:10.3390/ijerph17113993_

Round 1

Reviewer 1 Report

Sentence in line 57-59 doesn’t fit within the context of the paragraph.

Line 68- define MMR

Line 75- It isn’t apparent as to why there is a difference between outpatient and hospitalized pregnant women and self-medication. This may be due to cultural factors that are not obvious to those from other countries. For example, in the United States, pregnant women are hospitalized for medical reasons and they receive all of their medications from the hospital. If there are differences (do the women supply their own meds while hospitalized?), it would be helpful to explain.

Line 86-88- Is it important that the reader know the previous names of the hospital?

Study population- one of the wards was for abortions. Were women from this group included in the sample? They were not listed in the exclusion criteria but their decision to receive an abortion may have influenced their self-medication selection.

Line 159- for social drugs- were women asked about any other drugs besides smoking, alcohol, and khat chewing? If not, state why not and how maternal and fetal outcomes were targeted specifically for the three social drugs asked about. What are examples of medicinal plants that the women used?

Line 166- what was considered an adverse pregnancy outcome?

Line 218- what were some of the chronic diseases that the women had? This may impact pregnancy outcomes.

Were women asked how often, when, and how much of the social drugs were used during the pregnancy? This information is not clear and it is not explained.  For the amount of milliliters of alcohol consumed (table 4)- was this clarified by type of alcohol?

Were women asked at what term of their pregnancy the medications were used? Some drugs are more risky at certain points of the pregnancy.

Line 304- Did any of the women smoke at all during their pregnancy? You state none were “active smokers.” This may be related to poorer outcomes.

Line 333- put the info about recalling over a 9 month period at the beginning as this is important information to know

Line 362- the use of the phrase “prefer to self-care” appears presumptive as you early described decreased access to healthcare settings.

Is there a particular reason that women of Islam or Orthodox religions had higher use of medicinal plants? Perhaps add a sentence or two that may explain this (i.e. rituals, folk medicine, etc.).

You state that 1 in 10 women used an antibacterial drug but the table shows that a total of 20 out of the 300 women used an antibacterial drug. The math is off unless this information is coming from somewhere else.

You highlight that the most concerning aspect was the use of antibacterials used for self-medication. There were 20 women that used them, and all of the drugs listed were listed as “probably safe.” The use of khat was 65 out of 1117 (5.8%) and this drug was listed as having more dangerous side effects. The regular use of alcohol by the women was higher than both the khat and antibacterial but is not addressed in the discussion section. The risks of alcohol use during pregnancy are well known. It would be helpful to know what “medicinal plants” are in Ethiopia in order to understand the impact of 36.7% of the women using them. For example, in the United States, mint may be used for medicinal purposes but this plant has no adverse effects. Outlining this information would be helpful for readers not familiar with health care practices in Ethiopia.

Line 344- what is meant by “irrational medicine use”?

Overall, this study is very interesting and adds to the body of literature. The section on prescribed medications doesn’t need to be in this manuscript as the aim of the study was to examine the drugs the women used for self-medication.  The inclusion of adverse pregnancy outcomes is not relevant as there is not enough information to suggest a correlation. There are too many factors involved in an adverse outcome that are completely unrelated to the intake of medications. Your work is important and adds to the understanding of health and health care disparities.

Reviewer 2 Report

Dear Editors

This study is well written and carefully thought out. It needs minor amendments. 

 I have attached my comments .

Feedback on Pregnancy paper

The following need to be more clear as highlighted in the paper:

Page 1 line 17: need to be consistent with Jimma university medical center (JUMC) – Caps?

Page 2; line 84: again JUMC has already been abbreviated- so why not use abbrevs, only?

Page 3 line 94: MCH? What does this stand for?

Page 3 lines 107-113: I would like to have seen HOW this sample size calculation was carried out- was a reference used here to arrive to your sample size- it needs clarity 

Page 3 line 116: it would be more clearer to the reader about separate subheadings of inclusion /exclusion criteria. AS it stands, the reader is mean to assume anyone who is not excluded were included?

Page 3 line 120:& 126: maybe a personal thing to me but whilst data is plural, grammatically is it not better to say data WAS – I notice the authors have done this throughout the paper

Page 4: lines 135-139: in writing and translation of questionnaire from English to local dialects, was any sense of data collection compromised?

Page 4: lines 154: “assed”?  typo. However, these sentences are very confusing. The authors say “Section III

154  also assed medicinal plants use. Section IV dealt with prescription medicines, including

155  supplements, used among pregnant women prior to hospital admission.

Section III and IV, pregnancy

159  supplements were considered as medicines and data were collected about them.

My question is: Were supplements actually considered to be prescription only medicines or supplements were considered as medicines – if so why and how did authors arrive at their decision?  Again this needs absolute clarification as there is confusion to what constitutes medicines and food products?

I noticed section 2.5 line 171 – the authors have used UDFDA and AUTGA for classifying risk categories of medicines and it does make me wonder why these resources were not used to classify or identify supplements as food products or medicinal products? This is highly relevant to this study since what is viewed as a medicinal product in the Western world may only be viewed as a food product in LMIC and therefore used without  any safety concerns?

Supplements in their own rights can be considered as food products according to the UK MHRA https://www.gov.uk/guidance/decide-if-your-product-is-a-medicine-or-a-medical-device

Line 164: the use of the abstraction is rather misplaced here (subjective) since you are collecting data that is simply specific to your research surely?

Abstraction is a great word but misused here perhaps- once again a subjective comment on my part.

Lines 212- 213: results table 1: how is this calculated as it is rather confusing once again? Authors state “Around two thirds were from Oromo ethnic

213  group (69.7%), Muslims (65.4%),

HOWEVER, on examining the table 1 these figures seem mismatched especially where authors stae muslims (65.4%) and in the table 1 there are NO muslims but have ISLAM as a religion? Are muslims an ethnic minority (not mentioned in the table?) or is it in the context of a religion here ? more clarity needed.

Line 240: what is ATC? Line 257. – I noticed this is too far down the paper- needs to be where it is initially mentioned

Line 258-262: this is a huge statement to make. Can the authors explain /how they arrived to this outside of the “stats”? I am not convinced how much influence religion has on the self medication practice but education and social economic background combined with easy/cheaper access to healthcare would have an impact in LMIC.

Line 274: again the authors are stating “The most frequently prescribed group of medicines were supplemental medicines (97.5%). Ferrous sulphate was by far the single most prescribed medicine, (97.3%) (Table 3).

So now supplements are prescribed medicines? 

I would also point the authors in the direction of an excellent resource “drugs in pregnancy and lactating on medicines complete for an uptodate resource of various drugs and their effects in pregnancy mentioned in this paper?

Round 2

Reviewer 1 Report

I have reviewed the author's comments and revised manuscript.  I will include my comments below.   The manuscript is much improved in terms of flow and relevance of the information. I would like to thank the authors for their thoughtful and informative responses to my first review. As this is an international journal, it is helpful to understand local customs.   1- Information regarding adverse outcomes is referred to in the abstract and listed in the supplemental table. I recognize that this information was collected but it is, a) not defined as to what an adverse pregnancy outcome is, and b) not defined as to the relationship between the study aims and adverse outcome of the pregnancy. I appreciate that the authors defined adverse outcomes to me (this was not my intent as I am aware of them) but this is not clear to the reader and may not be related to intake of substances during the pregnancy. For example, a tight nuchal cord around the neck of the neonate or asynclitism may result in an adverse outcome but is unrelated to intake of substances during the pregnancy. The literature is clear and it is well known that substance use during pregnancy has an impact on the fetus. Your study was not designed to examine this variable in a manner that is useful and based on evidence. I would suggest removing any data or references to past and present adverse pregnancy outcomes other than in the background information where this relationship can be cited using appropriate studies.   2- The incorporation of the number of ml's for alcohol use is misleading. Local beer, beer, wine, vodka, and not given/indicated were listed as the types of alcohol consumed. The amount of daily intake was listed as "330 ml" as it was based on the description of a water glass. However, there is no distinction between 330ml of local beer and 330ml of vodka. There is clearly a difference in the amount consumed alcohol content between the two but this was not defined. Consumption of 330 mls of vodka scientifically has greater negative effects on the fetus that 330 mls  of beer. I would suggest removing the 330 mls and just keeping "regular or daily" use. Current recommendations is that alcohol not be consumed during pregnancy so the actual amount of alcohol does not change the recommendation. Removing this clarifier will be less confusing for the reader that has knowledge of maternal/fetal science.   Thank you for the work that was put into interviewing this many women. It would be interesting to note in the conclusions how this information will be used to continue on the downward trend of women self medicating during pregnancy.   Thank you,
